# Plant-Derived Nanoscale-Encapsulated Antioxidants for Oral and Topical Uses: A Brief Review

**DOI:** 10.3390/ijms23073638

**Published:** 2022-03-26

**Authors:** Seong-Hyeon Kim, Young-Chul Lee

**Affiliations:** Department of BioNano Technology, Gachon University, Seongnam-daero 1342, Sujeong-gu, Seongnam-si 13120, Korea; shaera@gachon.ac.kr

**Keywords:** nanomedicine, bio-active compounds, formulation

## Abstract

Several plant-based nanoscale-encapsulated antioxidant compounds (rutin, myricetin, β-carotene, fisetin, lycopene, quercetin, genkwanin, lutein, resveratrol, eucalyptol, kaempferol, glabridin, pinene, and whole-plant bio-active compounds) are briefly introduced in this paper, along with their characteristics. Antioxidants’ bioavailability has become one of the main research topics in bio-nanomedicine. Two low patient compliance drug delivery pathways (namely, the oral and topical delivery routes), are described in detail in this paper, for nanoscale colloidal systems and gel formulations. Both routes and/or formulations seek to improve bioavailability and maximize the drug agents’ efficiency. Some well-known compounds have been robustly studied, but many remain elusive. The objective of this review is to discuss recent studies and advantages of nanoscale formulations of plant-derived antioxidant compounds.

## 1. Introduction

Plant-derived antioxidants have many beneficial effects on humans; one of these is the reduction of the oxidative stress [1] that has been linked to the development of degenerative diseases and cancers [2,3,4]. In addition, many antioxidants exhibit anti-microbial effects [5], anti-inflammatory effects [6], and skin protection effects [7]. The present review focuses on the properties of carotenoids, polyphenols, terpenes, and xanthophylls (Table 1).

In the present review, we consider carotenoids such as β-carotene and lycopene. Carotenoids are synthesized not only by plants, but also by bacteria, fungi, and algae [56]. For example, carotenoids are present in salmon, tomato, and watermelon, explaining the colors of these foods [57,58]. In addition, carotenoids have been implicated in the mitigation of cardiovascular diseases, cancers [59], and osteoporosis [60,61]. Polyphenols, namely rutin, myricetin, fisetin, quercetin, genkwanin, resveratrol, kaempferol, and glabridin are discussed as well. It is well-known that, for example, chocolate [62] and green tea [63] are enriched in polyphenols. Polyphenols have been shown to mitigate carcinogenesis [64], cancers [65], and chronic diseases [66]. Terpenes, eucalyptol and pinene, are introduced in this review as penetration enhancers. Terpenes are commonly present in conifers [67], and act as skin-penetration enhancers by disrupting the lipid bilayer [68]. Terpenes have been reported to exhibit radical-scavenging effects [69] and anti-inflammatory effects [70,71,72]. Xanthophylls such as lutein are discussed as well. Xanthophylls are abundant in green vegetables and corn [73,74].

The effective delivery of drugs requires proper formulations. Drug properties (such as release kinetics, effectiveness, and retention time) are affected by specific formulations. Nanoscale miniaturization has been shown to improve the drugs’ bioavailability [75,76,77]. Owing to small particle sizes and high surface ratios, nanoscale formulations have become very desirable [78,79]. With the adoption of nanoscale formulations, the importance of nanoscale delivery systems has also increased [80,81]. The physical properties of nanoscale formulations (such as size, porosity, geometry, surface charge, and hydrophobicity) impart them with unique characteristics [82]. Nanoscale formulations can be used for effective targeting and delivery of drug agents, owing to the nanoparticles’ unique properties, which make them resilient to metabolic processes and immune response [78]. Several nanocolloid and nanogel formulations are discussed in this review. Colloidal systems are promising for various formulations, including ointments, emulsions, and aerosols. The synthesis of nanoparticles in colloidal systems has garnered significant attention. Gel (or nanogel) formulations have also been widely considered for synthesizing nanoparticles. Owing to the swelling effect of gels, they can be used for effectively delivering and/or controlling molecules [83,84].

The delivery route is one of the most addressed research issues in the nanomedicine field. Determining appropriate delivery routes is important for improving the drugs’ efficiency, which is often hindered owing to the barriers imposed by the human body. The oral and topical delivery routes are discussed in the present review. The oral delivery route is one of the more convenient routes but suffers from low patient compliance. Topical delivery, also among the more convenient drug delivery routes, allows the drugs to bypass the acidic environment of the stomach and the gastro-intestinal tract, but topically administered drugs have to overcome several hurdles in the skin layer [87]. Both routes (topical and oral) should ensure safe and effective drug delivery to the target. In this review, we discuss both the oral and topical routes for delivery of nanoscale colloidal and gel formulations (Figure 1).

## 2. The Oral Route

### 2.1. Gels

Mujtaba et al. studied nanogels containing rutin-loaded chitosan-alginate nanoparticles [88]. Rutin is present in passionflower, apples, and buckwheat [89]. Rutin exhibits anti-diabetic [8] and neuroprotective [9,10] effects. The absorption rate of rutin by the human body is poor, owing to the low water solubility of rutin [90]. The objective of the study by Mujtaba et al. was to formulate rutin-loaded nanoparticles with enhanced bioavailability and better water solubility and dissolution. Rutin-loaded chitosan-alginate nanoparticles were formulated using the coacervation method with slight modifications, and gels were formed using the ionotropic pre-gelation technique. Using the novel formulation, rutin was entrapped successfully by the nanogel particles. Rutin was released rapidly and sustainably from the rutin-loaded chitosan-alginate nanogel particles, whose positively charged surface enabled easy delivery of the payload to negatively charged cellular membrane channels [91].

Yao et al. studied chitosan-based nanogels of myricetin, for oral delivery [92]. Myricetin is abundantly found in berries and vegetables [93]. Myricetin was reported to inhibit cancers [16,17] and reduce hyperglycemia [18,19,20]. The nanogel formulation was prepared by following a typical procedure [92]. Nanogels with three-dimensional physical structures exhibited sustainable release properties [94], as well as good solubility and cellular uptake properties [95]. Myricetin-loaded nanogels exhibited the sustained and controlled release of myricetin, with good bioavailability.

Liu et al. studied β-carotene-encapsulated internal-phase emulsion gels, for edible applications, for improving the retention of β-carotene; for their studies, these authors simulated the gastro-intestinal tract conditions [96]. β-carotene is among the more well-known antioxidant components [21]. β-carotene-encapsulated internal-phase emulsion gels were formulated using a slightly modified cold gelation method [22]. Increasing the gel network density and antioxidant capacity improved the stability of β-carotene-loaded high-internal-phase emulsion gels. In addition, the improved bio-accessibility of β-carotene was demonstrated for β-carotene-loaded gels and was attributed to their gel network structure.

Whole flavonoids of Satsuma mandarin (*Citrus unshiu*) peel extracts with pectin nanoparticles were considered by Hu et al. for effective delivery [97]. Before loading with pectin nanoparticles, Satsuma mandarin peels were extracted, and whole components of Satsuma mandarin were used in the studies. Then, the Satsuma mandarin peel extract was loaded with pectin nanoparticles, using the gelation technique [98]. Owing to the novel formulation, Satsuma mandarin peel gels exhibited improved bio-accessibility and controlled release of bioactive compounds.

Mahmoudi et al. formulated gels with chitosan nanoparticles containing the Chinese lantern (*Physalis alkekengi* L.) extract [99]. The Chinese lantern extract contains bioactive compounds, including carotenoids [100] and flavonoids [101]. This extract has been used in traditional medicine [102]. The seeds of the Chinese lantern were dried, ground, and then extracted using the percolation method. Next, the Chinese lantern extract was entrapped in chitosan nanoparticles using the ionic gelation method [103]. Polycationic chitosan presents high bio-affinity to a negative charged cell surface, including a negative charged cancer cell or bacteria membrane [104,105]. Using this formulation, high-biocompatibility oral consumption became possible. Chitosan nanoparticles containing the Chinese lantern extract exhibited good antioxidant effects and stability, enabling the protection of bioactive compounds using the novel formulation.

### 2.2. Colloids

Sechi et al. studied the encapsulation of fisetin with polymeric nanoparticles, for controlled oral delivery and release (Figure 2) [106]. Polymers in this work were biodegradable and biocompatible, of the type that is typically used for the encapsulation of drugs to enhance oral bioavailability. Fisetin, commonly found in fruits and vegetables, such as onions and strawberries [107], was reported as an antioxidant flavonoid [108,109] with anti-cancer effects [23,24]. Fisetin was loaded into nanoparticles, using a modified nanoprecipitation method [110]. With polymeric nanoparticles, fisetin-loaded nanoparticles exhibited an effective fisetin-loading capacity and controlled release of fisetin, in simulated gastro-intestinal conditions.

Regiellen et al. studied rutin-loaded bovine serum albumin nanoparticles for addressing the low oral bioavailability of rutin [11]; the study was performed using nanoscale spray drying methods [12]. Rutin-loaded bovine serum albumin nanoparticles exhibited improved radical scavenging after 72 h, implying a sustained release of rutin. Rutin penetrated into the tissue, owing to the fine particle size of rutin-loaded nanoparticles, yielding better performance than original drugs for similar in vivo conditions.

Singh et al. considered a lipid-based lycopene nanoemulsion system for improving oral bioavailability [111]. Lycopene, a carotenoid, is found in tomatoes [112] and olives [113]. It is well-known that lycopene inhibits and prevents prostate cancer [25,26]. Singh et al. formulated lycopene-loaded nano-lipid carriers, using the ultra-sonication method. Lipid nanoparticles participated in lipophilicity of the drug, so that the drug entered easily to the central nerve system (CNS) [114]. Lycopene-loaded nano-lipid carriers exhibited no precipitates or phase separation, yielding high dispersibility. Lipid nanoparticles delivered the drug inside of the cells for nucleoside transporters so the drug could access rapidly with lycopene into the cells. High stability of this formulation, good gut permeation, and good cytotoxicity against human breast tumor cells were shown for lycopene-loaded nano-lipid carriers.

Hao et al. considered quercetin-loaded polymeric nanocapsules with soybean lecithin [115]. Quercetin is a flavonoid that is abundantly present in onions, berries, and apples [116]. Quercetin was reported to inhibit the progression of the cancer cell cycle [28,29,30], and to downregulate cardiovascular diseases [31]. Hao et al. used the electrostatic deposition method for encapsulating quercetin with chitosan-coated liposomes. Quercetin-loaded nanoparticles in an aqueous solution exhibited higher stability, reducing power, and cytotoxicity with respect to the human liver hepatocellular carcinoma cells compared with free quercetin. Trypan blue staining in the MTT assay revealed cytotoxicity to the human liver hepatocellular carcinoma cells by observation of blue spots inside of the cells.

Li. et al. considered a nanosuspension system with genkwanin [117]. Genkwanin is found in the seeds of black alder (*Alnus glutinosa*) [117] and laurel-leaf cistus (*Cistus laurifolius*) [118]. Genkwanin has been reported to inhibit breast cancer [34,119]. A nanosuspension system containing genkwanin was synthesized using the anti-solvent precipitation method [35]. Genkwanin nanosuspensions demonstrated a stronger anti-tumor effect and tolerance, enabling the oral delivery route.

Zhou et al. studied whey protein isolate-based nanoemulsions for the bio-accessibility of edible β-carotene [120]. Interestingly, Zhou et al. used whole edible ingredients in their formulation. The bio-accessibility and stability of β-carotene increased owing to the smaller particle size, suggesting a suitable formulation of this oil-based bioactive compound.

For management of osteoporosis, Gera et al. developed a rutin-nanoparticle colloidal system [121] using the anti-solvent precipitation technique [13]. The rutin nanosuspension exhibited a high drug absorption rate, good solubility, and good intestinal permeability. Enhanced bioavailability, dose reduction, and long-term stability were also demonstrated by this novel formulation.

Kumar et al. formulated isradipine-loaded solid lipid nanoparticles with rutin [122]. Solid lipid nanoparticles were also formulated using a method developed by Gardouh et al. [123]. Isradipine was shown to treat stroke and heart attacks by blocking calcium channels [124]. Isradipine released from the formulated solid lipid nanoparticles exhibited enhanced sustainability and a higher absorption rate than original isradipine, owing to rutin.

For improving the oral bioavailability of lycopene, Mishra and Kumari developed lycopene nanosuspensions [75]. The nanoprecipitation method was used for formulating a lycopene nanosuspension system [27]. Lycopene nanosuspensions decreased the level of triglycerides; at the same time, they improved the effect of insulin. It was also shown that lycopene nanosuspensions increased the amount of the released drug and reduced the blood glucose level.

## 3. Topical Use

### 3.1. Gels

To develop topical delivery through the eye, Bodoki et al. considered a thermosensitive lutein nanogel system for treating selenite-induced cataracts [125]. Lutein is present in kale [126] and spinach [127,128]. Lutein has been widely shown to exert beneficial effects on eyes [36,37]. Lutein-loaded poly(lactic-co-glycolic acid) (PLGA) nanogels were formulated using the emulsion/evaporation method, while lutein-loaded zein nanogels were formulated using a liquid-liquid dispersion. Lutein-loaded nanogels demonstrated better stability and more efficient delivery to the lens, compared with the similar amounts of free lutein. These nanogels were shown to reduce selenite-induced cataracts, with better bioavailability and longer antioxidant retention.

Andleeb et al. developed yarrow (*Achillea millefolium*) extract-loaded nanoethosomes for topical delivery through skin (Figure 3) [129]. Yarrow is well-known as a traditional medical plant [130]. The yarrow extract reportedly exerts choleretic effects [130,131]. Andleeb et al. used a simple cold method for loading the yarrow extract into nanoethosomes. In the case of dermal delivery, ethosomes cause skin disruption. More and deeper ethosomes could permeate inside of the skin [132]. By loading the yarrow extract into nanoethosomes, ethanol helped to penetrate easily through the skin. In addition, with a narrow distribution of particle sizes, small-size yarrow extract-loaded nanoethosomes successfully delivered bioactive compounds into deep skin layers. These unique properties imparted the nanoethosomes with better skin penetration characteristics than common gels [133,134], demonstrating a higher efficiency of the yarrow extract.

Imran et al. studied a nanostructured lipid carrier gel for loading two drugs, quercetin and resveratrol, to improve the disposition for their topical delivery [135]. Resveratrol is a polyphenol compound found in grapes, peanuts [136], mulberry fruit, and Jamun seed [137] and was shown to exhibit anti-cancer effects [38,39,40] and the reduction of cardiovascular risk factors [41,42], which explains the beneficial effects of the wine [138]. Nanostructured lipid carriers were prepared by melt emulsification using the ultra-sonication method and formulated to the gel form using the methods of Naz et al. [139]. The nanostructured lipid carrier gel exhibited better skin hydration, owing to its nanoscale particles with the novel formulation, leading to better permeation of drugs encapsulated in the gel particles. The nanostructured lipid carrier gel demonstrated lower IC_50_ and better drug permeation than the conventional gel, implying the better topical bioavailability of the former. In addition, the nanostructured lipid carrier gel demonstrated the inhibition of migration in a bidirectional wound-healing assay, owing to its anti-metastatic effect.

Gokhale et al. developed a gel based on a quercetin-loaded nanoemulsion [140]. Quercetin was used as a drug agent for treating rheumatism, while the nanoemulsion formulation was used for effective topical delivery. The formulation was made using the spontaneous emulsification technique. Then, gels were synthesized based on the prepared nanoemulsion [141]. Quercetin-loaded gels exhibited good solubility, diffusion rate, skin permeability, and physicochemical stability.

Rutin nanocrystal gels were studied by Li et al. for improving the bioavailability and efficiency of rutin [142]. Rutin nanosuspensions were formulated using high-speed shearing and the high-pressure homogenization technique [14,15]. After preparing rutin nanosuspensions, they were freeze-dried [143] and dispersed to form a nanocrystal gel. There was a remarkable improvement of the saturation solubility, release behavior, transdermal bioavailability of the drug, and antioxidant activity of the rutin nanocrystal gel. In addition, the nanocrystal gel inhibited the oxidative damage associated with the skin photoaging.

### 3.2. Colloids

Hatahet et al. formulated a quercetin smartCrystals^®^ into a nanosuspension system [144]. As mentioned above, quercetin has some beneficial health effects; on the other hand, its bioavailability is poor. By formulating quercetin into nanocrystals, its bioavailability can be improved, owing to the nanocrystals’ high performance when delivered through the dermal route [145]. Hatahet et al. synthesized quercetin nanosuspensions, using the smartCrystals^®^ technology [32]. Using this novel formulation, quercetin nanosuspensions exhibited higher saturation solubility, better antioxidant activity, and better physical stability. In addition, the protective activity on Vero cells with respect to hydrogen peroxide toxicity was demonstrated and the MTT assay revealed that quercetin smartCrystals^®^ do not show cellular cytotoxicity at a higher concentration than crude quercetin.

To improve the delivery of drugs and large-molecular-weight compounds into deep skin layers, Kahraman et al. formulated nanomicelles with a combination of terpenes and tacrolimus, obtaining an aqueous formulation for topical delivery [146]. Tacrolimus is an immunosuppressive drug that is used in transplant medicine [147,148]. Comprehensive terpenes were used for enhancing the tacrolimus penetration for topical delivery. Terpenes are powerful skin-penetration enhancers [149,150] and anti-inflammation agents [151,152]. Tacrolimus monohydrate-loaded polymeric micelles were prepared using the thin film hydration method [153]. Nanomicelles enhanced the drug delivery through skin [154,155]. Owing to this novel formulation, tacrolimus-loaded nanomicelles exhibited improved colloidal stability, higher drug-loading efficiency, and higher accumulation of large molecules in the viable epidermis and dermis.

Chao et al. studied a kaempferol-loaded nanoemulsion system for topical delivery [156]. Kaempferol has poor water solubility, which implies a low bio-absorption rate [157]; thus, a kaempferol-loaded nanoemulsion system has been considered for overcoming this problem. Kaempferol is abundantly found in broccoli, spinach, beans, and kale [48]. Importantly, it was reported that kaempferol reduces the risk of Alzheimer’s disease [48,49,50]. The kaempferol-loaded nanoemulsion system exhibited enhanced permeation capacity, a higher drug accumulation over the period of 12 h, a higher deposition amount in skin, and a lower lagging time.

Using glabridin, Wang et al. designed a nanosuspension system for topical delivery [158]. Glabridin, a flavonoid, is typically found in licorice [159,160], and was reported to inhibit the tyrosinase activity [51]. Wang et al. formulated a glabridin nanosuspension system using the anti-solvent precipitation-homogenization method [52]. Using this method, Wang et al. demonstrated that the glabridin nanosuspension system is promising for topical use. Glabridin nanosuspension has higher solubility due to its reduced size, so that several advantages are demonstrated. A specially formulated glabridin nanosuspension system demonstrated enhanced stability for short-term storage, with no significant particle aggregation; it also enhanced skin permeation, thanks to enhanced skin penetration due to an enhanced concentration gradient, both in vitro and in vivo.

Nikolic et al. studied curcumin-loaded nanoemulsions containing eucalyptol and pinene, for evaluating a curcumin-loaded low-energy nanoemulsion containing terpenes, for curcumin delivery [161]. Curcumin, a plant-derived polyphenol, is used in skin disease treatments [162,163]. Terpenes, eucalyptol, and pinene were used as penetration enhancers. Eucalyptol was reported to exhibit anti-inflammatory [43,44] and lung-protective effects [45,46,47]. Pinene is a major component of the essential oil in sage [163] and was reported to exhibit anti-tumor activity [53,54,55]. The spontaneous emulsification method was used for formulating the nanoemulsion systems. Both eucalyptol and pinene acted as surfactants, imparting the nanoemulsions with a low interface energy, and reduced the amount of the surfactant needed for maintaining a stable formulation, which resulted in higher safety with enhanced penetration effects of the nanoemulsions.

In another study, solid lipid nanoparticles were formulated with a mixture of surfactants [164]. Bose et al. developed a solvent-free solid lipid-based nanoscale system for topical delivery. Quercetin was used as a drug agent, and quercetin nanoparticles were prepared using the probe ultra-sonication method [33]. Owing to the drug enrichment in the outer shell, the diffusion path of the active component was shorter. Remarkable physical stability and a high initial burst-like release, as well as a prolonged release, were observed for quercetin-encapsulated solid lipid nanoparticles.

## 4. Conclusions

In this article, we reviewed some recent studies on plant-derived antioxidants. Applications of plant-derived antioxidants have been widely discussed. Several plant-derived antioxidants were reviewed in this paper, but the discussion is by no means exhaustive, implying additional discoveries. In addition, while some components have been well-characterized, others remain elusive.

Much remains to be discovered in the field of nanomedicine. The unique properties of nanoscale compounds are notable but remain elusive, implying the need for additional research and optimization [165,166]. At the same time, nanomedicine approaches are very promising for improving the efficiency of drug delivery systems, which in turn is likely to improve the quality of life.

## Figures and Tables

**Figure 1 ijms-23-03638-f001:**
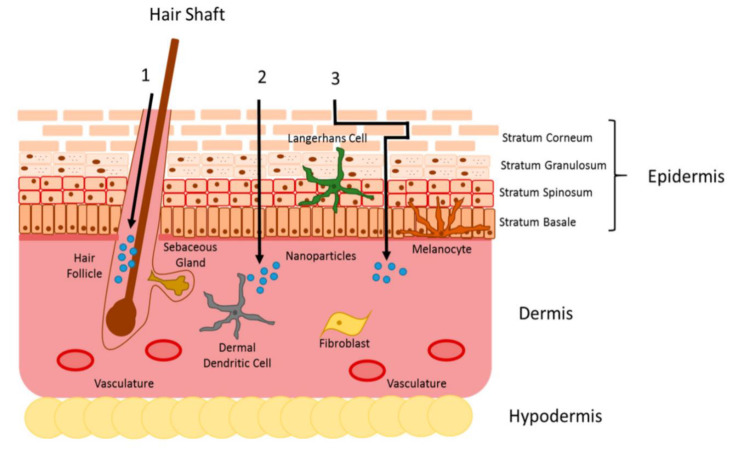
Skin penetration delivery route and physiological factors affecting the oral delivery route in the gastrointestinal tract. Reprinted from Nanomedicine: Nanotechnology, Biology and Medicine, 11./5, Susan Hua et al., Advances in oral nano-delivery systems for colon targeted drug delivery in inflammatory bowel disease: Selective targeting to diseased versus healthy tissue, 1117–1132. Copyright (2015), with permission from Elsevier [85] and MDPI [86].

**Figure 2 ijms-23-03638-f002:**
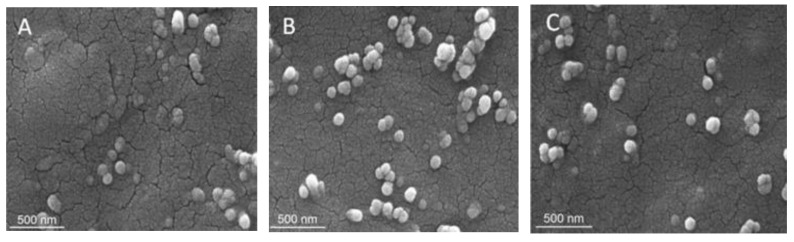
(**A**–**C**) SEM images of fisetin nanoparticles [106]. Adopted from Sechi, Mario, et al., “Nanoencapsulation of dietary flavonoid fisetin: Formulation and in vitro antioxidant and α-glucosidase inhibition activities”. *Materials Science and Engineering: C* 68 (2016): 594–602, with permission of Elsevier.

**Figure 3 ijms-23-03638-f003:**
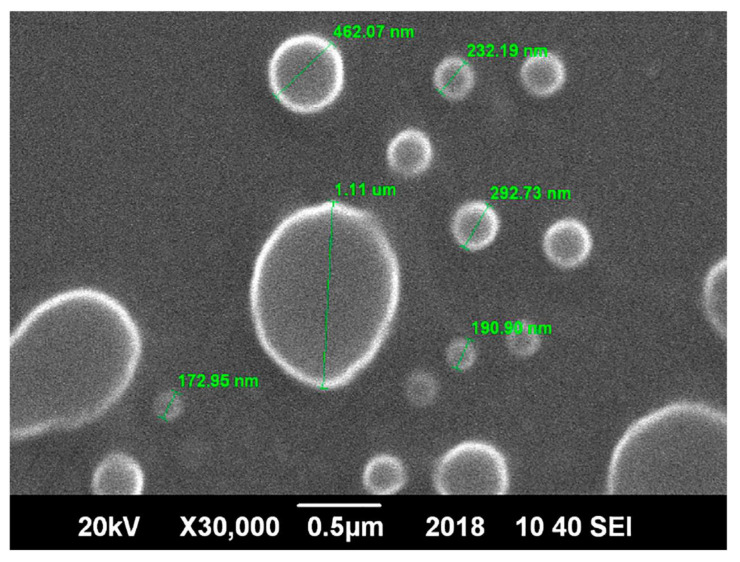
SEM image of the herbal nanoethosomes [129]. Adapted from *Andleeb* et al., “Development, Characterization and Stability Evaluation of Topical Gel Loaded with Ethosomes Containing Achillea millefolium L. Extract”. *Frontiers in pharmacology* 12 (2021) 336, with permission of Frontiers.

**Table 1 ijms-23-03638-t001:** Classification, effects, and nanonization strategies for plant-derived antioxidants considered in this paper.

Name	Structure	Classification	Effect	Nanonization Strategy
Rutin	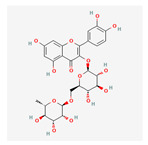	Polyphenol	Anti-diabetic effects [8] Neuroprotective effects [9,10]	Coacervation method Nano-spray drying methods [11,12] Anti-solvent precipitation technique [13] High-speed shearing and high-pressure homogenization technique [14,15]
Myricetin	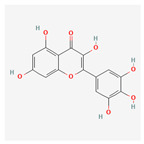	Polyphenol	Anti-cancer effects [16,17] Hyperglycemia reduction [18,19,20]	Not considered in this review
β-carotene	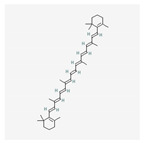	Carotenoid	Radical scavengers [21]	Cold gelation method [22]
Fisetin	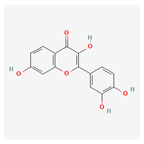	Polyphenol	Anti-cancer effects [23,24]	Nanoprecipitation method
Lycopene	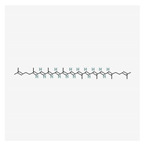	Carotenoid	Inhibition and prevention of prostate cancer [25,26]	Ultrasonication method Nanoprecipitation technique [27]
Quercetin	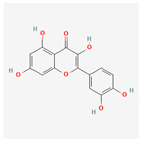	Polyphenol	Inhibition of the cancer cell cycle progression [28,29,30] Regulation of cardiovascular disease [31]	Electrostatic deposition Spontaneous emulsification technique SmartCrystals^®^ technology [32] Probe ultra-sonication method [33]
Genkwanin	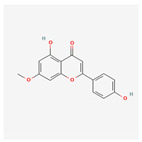	Polyphenol	Inhibition of breast cancer [34]	Anti-solvent precipitation method [35]
Lutein	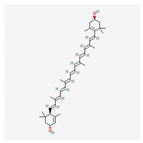	Xanthophyll	Beneficial eye effects [36,37]	Emulsion/evaporation method Liquid-liquid dispersion
Resveratrol	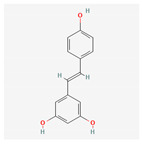	Polyphenol	Anti-cancer effects [38,39,40] Reduction of cardiovascular risk [41,42]	Melt emulsification with ultra-sonication method
Eucalyptol	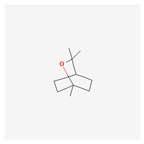	Terpene	Anti-inflammatory effects [43,44] Lung protective effects [45,46,47]	Spontaneous emulsification method
Kaempferol	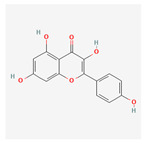	Polyphenol	Reduction of Alzheimer’s disease risk [48,49,50]	Not considered in this review
Glabridin	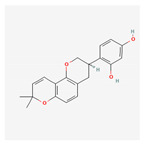	Polyphenol	Inhibition of tyrosinase activity [51]	Anti-solvent precipitation-homogenization method [52]
Pinene	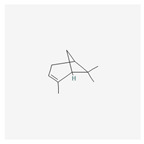	Terpene	Anti-cancer activity [53,54,55]	Spontaneous emulsification method

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
