# Peer review of "Plant-Derived Nanoscale-Encapsulated Antioxidants for Oral and Topical Uses: A Brief Review"

_ijms, 2022, doi:10.3390/ijms23073638_

Round 1

Reviewer 1 Report

While the paper by Kim and Lee is a good brief review of nano gels and colloids, some more details could be added before publication. 

1) Figure of all mentioned bioactive small molecules should be added, together with some general figures of nano gels and nano colloids structures, so that the readers could have a better image about the topic of the paper.  

2) Some more details about the mentioned nano materials, like are they in cited papers formulated to target specific tissue or not. If they are used to target specific tissues, how was this done? Also from the mentioned results, what is obtained on cell lines and what in vivo? Some more details about the mentioned procedures for the making of nano gels and colloids can be added as well.

3) Regarding oral application, was the first pass effect mentioned in any of the cited papers. If so, what is the effect of it on the bioavailability of studied nano materials? If this was not discussed previously, some discussion or the authors opinion about it should be added.

4) Line 195. There is a mention that resveratrol is found in grapes and peanuts. While this is correct, this molecule is found in many more natural sources and also some man made, like wine for example. Please extend this.

5) Is there so far any commercially available product based on techniques mentioned in this paper? If there is, it should be mentioned.

Author Response

Question 1)

Figure of all mentioned bioactive small molecules should be added, together with some general figures of nano gels and nano colloids structures, so that the readers could have a better image about the topic of the paper. 

Answer 1.

Thank you for your valuable comments. Figures of all mentioned bioactive small molecules have been added in the revised ms. Molecular structure Figures are adopted from pubchem, where we can reuse in our own publication without any special permission*; (Page 6-8 on Line 296, Table 1.)

*https://pubchemdocs.ncbi.nlm.nih.gov/citation-guidelines$_2-3

General figures of nano gels and nano colloids structures are also added with permission of Frontiers and Elsevier. We hope these figures which adapted from reviewed paper would help readers to understand structures of nano gels and nano colloids; (Page31 on Line 738-740, Line741-743)

Question 2)

Some more details about the mentioned nano materials, like are they in cited papers formulated to target specific tissue or not. If they are used to target specific tissues, how was this done? Also from the mentioned results, what is obtained on cell lines and what in vivo? Some more details about the mentioned procedures for the making of nano gels and colloids can be added as well.

Answer 2.

More details regarding to targeting specific tissue, results that obtained on cell lines or in vivo are added. Details about the mentioned procedures are written shortly because we thought too much of details can disrupt the readability of this paper. Readers would find details in the references.

Line 110-112 in page 3

Seeds of Chinese lantern were dried, grinded, and then extracted using the percolation method. Next, the Chinese lantern extract was entrapped in chitosan nanoparticles using the ionic gelation method. Polycationic chitosan presents high bio-affinity to negative charged cell surface, including negative charged cancer cell or bacteria membrane. Using this formulation, high-biocompatibility oral consumption became possible. Chitosan nanoparticles containing the Chinese lantern extract exhibited good antioxidant effects and stability, enabling the protection of bioactive compounds using the novel formulation.

Line 135-140 in page 3

Singh et al. formulated lycopene-loaded nano-lipid carriers, using the ultra-sonication method. Lipid nanoparticles participate in lipophilicity of the drug, so that the drug enters easily to the Central Nerve System (CNS). Lycopene-loaded nano-lipid carriers exhibited no precipitates or phase separation, yielding high dispersibility. Lipid nanoparticles deliver the drug inside of the cells for nucleoside transporters so the drug can access rapidly with lycopene into the cells. High stability of this formulation, good gut permeation, and good cytotoxicity against human breast tumor cells were shown for lycopene-loaded nano-lipid carriers.

150-151 in page 4

Hao et al. used the electrostatic deposition method for encapsulating quercetin with chitosan-coated liposomes. Quercetin-loaded nanoparticles in an aqueous solution exhibited higher stability, reducing power, and cytotoxicity with respect to the human liver hepatocellular carcinoma cells compared with free quercetin. Trypan blue staining in MTT assay revealed cytotoxicity to the human liver hepatocellular carcinoma cells by observation of blue spots inside of the cells.

Line 192-194 in page 4

Andleeb et al. used a simple cold method for loading the yarrow extract into nanoethosomes. In case of dermal delivery, ethosomes cause skin disruption. More and deeper ethosomes could permeate inside of the skin. By loading the yarrow extract into nanoethosomes, ethanol helped to penetrate easily through the skin. In addition, with a narrow distribution of particle sizes, small-size yarrow extract-loaded nanoethosomes successfully delivered bioactive compounds into deep skin layers.

Line 236-237 in page 5

Using this novel formulation, quercetin nanosuspensions exhibited higher saturation solubility, better antioxidant activity, and better physical stability. In addition, protective activity on Vero cells with respect to hydrogen peroxide toxicity was demonstrated and MTT assay revealed that quercetin smartCrystals® do not show cellular cytotoxicity at higher concentration than crude quercetin.

Line 262-263 in page 6

Using this method, Wang et al. demonstrated that the glabridin nanosuspension system is promising for topical use. Glabridin nanosuspension has higher solubility due to its reduced size, so that several advantages are demonstrated. A specially formulated glabridin nanosuspension system demonstrated enhanced stability for short-term storage, with no significant particle aggregation;

Line 265-266 in page 6

A specially formulated glabridin nanosuspension system demonstrated enhanced stability for short-term storage, with no significant particle aggregation; it also enhanced skin permeation, thanks to enhanced skin penetration due to enhanced concentration gradient, both in vitro and in vivo.

Question 3)

Regarding oral application, was the first pass effect mentioned in any of the cited papers. If so, what is the effect of it on the bioavailability of studied nano materials? If this was not discussed previously, some discussion or the authors opinion about it should be added.

Answer 3.

We checked all the papers that reviewed in this paper but there are no any paper which mentioned first pass effect Most of papers discussed bioavailability regarding to gastrointestinal simulation.

Question 4)

Line 195. There is a mention that resveratrol is found in grapes and peanuts. While this is correct, this molecule is found in many more natural sources and also some man made, like wine for example. Please extend this.

Answer 4.

We added 2 more natural sources of resveratrol on line 202. Mulberry fruit and Jamun seed also contains the resveratrol and analysis work of mulberry fruit and jamun is done abundantly. We added mulberry fruit and jamun for more natural sources because we think it is well defined.

Regarding to wine, we added mention on line 204 as an example of resveratrol’s useful effect.

Line 202-204

, mulberry fruit, and Jamun seed and was shown to exhibit anti-cancer effects and reduction of cardiovascular risk factors, which explains beneficial effects of the wine. Nanostructured lipid carriers were prepared by melt emulsification using the ultra-sonication method and formulated to the gel form using the methods of Naz et al.

Question 5)

Is there so far any commercially available product based on techniques mentioned in this paper? If there is, it should be mentioned.

Answer 5.

We hoped to find and add more details, but we still could not find commercially available product based on techniques mentioned in this paper. Producers don’t reveal their formulation technique or recipe, and we do think many of techniques handled in this paper are still in experimental progress. We are sorry that we could not mention the explanation.

One more thing, we found a word that spell should be corrected on line 228 on page5.

smartrystal -> smartCrystals®

Reviewer 2 Report

Seong-Hyeon Kim and Young-Chul Lee wrote an interesting paper on antioxidants in nanoscopic capsules for oral and topical use. The paper contains rich literature, but most references are from 2000-2016. According to the reviewer, when writing an up-to-date review, we should mainly focus on sources from the last three years. On a similar topic, reviews such as doi:10.3390/antiox9100923, doi:10.3389/fbioe.2019.00447, doi:10.1007/s42452-020-3110-8 can be found. Therefore, I believe that the article is not suitable for publication in its current form. I also encourage the authors to make more tables, figures, etc. 

Author Response

Seong-Hyeon Kim and Young-Chul Lee wrote an interesting paper on antioxidants in nanoscopic capsules for oral and topical use. The paper contains rich literature, but most references are from 2000-2016. According to the reviewer, when writing an up-to-date review, we should mainly focus on sources from the last three years. On a similar topic, reviews such as doi:10.3390/antiox9100923, doi:10.3389/fbioe.2019.00447, doi:10.1007/s42452-020-3110-8 can be found. Therefore, I believe that the article is not suitable for publication in its current form. I also encourage the authors to make more tables, figures, etc. 

Answer

We thank for your valuable comments. Most of papers we reviewed in this paper are written after 2016. Only 2 papers are written before 2016, since these papers include remarkable experiments. Actual reviewed papers are written in last 6 years, so that we believe we can describe this paper as recent review. Following list is papers reviewed in this paper and published year.

Chitosan-Alginate Nanoparticles as a Novel Drug Delivery System for Rutin : 2018(page2)

Preparation and evaluation of chitosan-based nanogels/gels for oral delivery of myricetin : 2016(page2)

Stability, rheology, and β-carotene bioaccessibility of high internal phase emulsion gels : 2019(page2)

Chitosan nanoparticles containing Physalis alkekengi-L extract: preparation, optimization and their antioxidant activity : 2019(page3)

In vitro release and antioxidant activity of Satsuma mandarin (Citrus reticulata Blanco cv. unshiu) peel flavonoids encapsulated by pectin nanoparticles : 2017(page3)

Nanoencapsulation of dietary flavonoid fisetin: Formulation and in vitro antioxidant and alpha-glucosidase inhibition activities : 2016(page3)

Bovine serum albumin-based nanoparticles containing the flavonoid rutin produced by nano spray drying : 2020(page3)

Lipid Based nanoformulation of lycopene improves oral delivery: formulation optimization, ex vivo assessment and its efficacy against breast cancer : 2017(page3)

Encapsulation of the flavonoid quercetin with chitosan-coated nano-liposomes : 2017(page3)

Genkwanin nanosuspensions: a novel and potential antitumor drug in breast carcinoma therapy : 2017(page4)

Stability and in vitro digestibility of beta-carotene in nanoemulsions fabricated with different carrier oils : 2018(page4)

Rutin nanosuspension for potential management of osteoporosis: effect of particle size reduction on oral bioavailability, in vitro and in vivo activity : 2020(page4)

Optimization & design of isradipine loaded solid lipid nanobioparticles using rutin by Taguchi methodology : 2016(page4)

Engineering of Crystalline Nano-Suspension of Lycopene for Potential Management of Oxidative Stress–Linked Diabetes in Experimental Animals : 2021(page4)

Topical nanodelivery system of lutein for the prevention of selenite-induced cataract : 2019(page4)

Development, Characterization and Stability Evaluation of Topical Gel Loaded With Ethosomes Containing Achillea millefolium L. Extract : 2021(page4)

Topical nanostructured lipid carrier gel of quercetin and resveratrol: Formulation, optimization, in vitro and ex vivo study for the treatment of skin cancer : 2020(page5)

Quercetin loaded nanoemulsion-based gel for rheumatoid arthritis: In vivo and in vitro studies : 2019(page5)

A rutin nanocrystal gel as an effective dermal delivery system for enhanced anti-photoaging application:2021(page5)

Dermal quercetin smartCrystals(R): Formulation development, antioxidant activity and cellular safety : 2016(page5)

The combination of nanomicelles with terpenes for enhancement of skin drug delivery : 2018(page5)

The Effect of Submicron Emulsion Systems on Transdermal Delivery of Kaempferol : 2012(page5)

Glabridin nanosuspension for enhanced skin penetration: formulation optimization, in vitro and in vivo evaluation : 2016(page6)

Microstructure and biopharmaceutical performances of curcumin-loaded low-energy nanoemulsions containing eucalyptol and pinene: Terpenes’ role overcome penetration enhancement effect? : 2020(page6)

Formulation optimization and topical delivery of quercetin from solid lipid based nanosystems : 2013(page6)

One more thing, we found a word that spell should be corrected on line 228 on page5.

smartrystal -> smartCrystals®

Round 2

Reviewer 1 Report

The authors have improved the paper significantly and addressed all of my issues. I recommend its publication. 

Reviewer 2 Report

The current version is better than the previous one, so I recommend publishing the manuscript in the current form.